# Efficient Utilization of Fruit Peels for the Bioproduction of D-Allulose and D-Mannitol

**DOI:** 10.3390/foods11223613

**Published:** 2022-11-12

**Authors:** Jin Li, Jiajun Chen, Wei Xu, Wenli Zhang, Yeming Chen, Wanmeng Mu

**Affiliations:** 1State Key Laboratory of Food Science and Technology, Jiangnan University, Wuxi 214122, China; 2School of Food Science and Technology, Jiangnan University, Wuxi 214122, China; 3International Joint Laboratory on Food Safety, Jiangnan University, Wuxi 214122, China

**Keywords:** fruit peel, Ketose 3-epimerase, D-Mannitol 2-dehydrogenase, D-Allulose, D-Mannitol

## Abstract

Currently, the demand for low-calorie sweeteners has grown dramatically because consumers are more mindful of their health than they used to be. Therefore, bioproduction of low-calorie sweeteners from low-cost raw materials becomes a hot spot. In this study, a two-stage strategy was established to efficiently utilize D-fructose from fruit and vegetable wastes. Firstly, ketose 3-epimerase was used to produce D-allulose from D-fructose of pear peels. Secondly, the residual D-fructose was converted to D-mannitol by the engineered strain co-expression of D-mannitol 2-dehydrogenase and formate dehydrogenase. Approximately 29.4% D-fructose of pear peels was converted to D-allulose. Subsequently, under optimal conditions (35 °C, pH 6.5, 1 mM Mn^2+^, 2 g/L dry cells), almost all the residual D-fructose was transformed into D-mannitol with a 93.5% conversion rate. Eventually, from 1 kg fresh pear peel, it could produce 10.8 g of D-allulose and 24.6 g of D-mannitol. This bioprocess strategy provides a vital method to biosynthesize high-value functional sugars from low-cost biomass.

## 1. Introduction

With the improvement of economic life, the demand and consumption of sugars have increased globally over the past decades. In the World Health Organization’s (WHO) *World Health Statistics Report 2021*, the age-standardized mortality rate for diabetes increased by 3%, and the rate of increase in diabetes deaths was closely related to the increase in obesity rates. Meanwhile, more than 1.9 billion adults worldwide were overweight, and more than 650 millions of them were obese, accounting for 13% of the world’s total adult population [1]. The WHO highly advised keeping free sugar intake to less than 10% of total caloric intake due to the public health problems associated with high sugar consumption. In terms of sugar control, more and more countries have implemented sugar taxes in recent years, and sugar reduction has become an important trend in food production and consumption. It is an effective strategy to use alternative sweeteners to substitute for sugars, particularly in foods, drinks and medicines, which are likely to be consumed between meals [2]. 

D-Allulose is one of the most promising low-calorie rare sugars, and it is the epimer of D-fructose at the C-3 position. There are several distinct physiological properties of D-allulose, including modest improvements in postprandial glucose and insulin regulation, [3], anti-obesity and prevention of type 2 diabetes [4,5], anti-hyperlipidemia, anti-hyperglycemic effects [6], anti-inflammatory [7], and anti-atherosclerosis effects [8]. It also prevents diabetic nephropathy [9], enhances endurance ability, reduces fatigue [10,11], and inhibits hyperphagia with excessive appetite [12]. Moreover, it contains a sweetness of 70% sucrose but only 0.4 kcal/g of calories. In addition to being utilized as a low-calorie sweetener, D-allulose can also improve texture and antioxidant activity, increase the protein emulsification and foaming abilities of bakery [13], and enhance the viscosity, elasticity and water holding capacity of frozen foods to increase storage stability [14]. Through C-3 epimerization, D-allulose can be synthesized from D-fructose [15]. In this process, ketose 3-epimerase (KEase) plays a vital role [16]. Almost 30 types of KEase have been identified and described from various microorganisms so far [17]. It is known that about 30% of D-fructose gets converted into D-allulose, however, which is the major restriction of industrial production of D-allulose. In comparison, the theoretical conversion rate of D-mannitol enzymatic production from D-fructose could reach 100%. This provides a solution to sufficiently utilize D-fructose for the bioproduction of value-added saccharides. 

D-Mannitol is a naturally occurring, low-calorie alditol with various applications in the food and pharmaceutical industries. In the food industry, D-mannitol usually has utility as a functional food due to its low metabolic rate and no glycemic index [18], and it is widely used as a sugar-free coating because of its non-hygroscopic properties [19]. Furthermore, in the pharmaceutical industry, by activating superoxide dismutase and reducing the accumulation of lipid peroxides in the body, D-mannitol has a prospective utility for removing free radicals [20]. Due to its potent dehydrating and osmotic diuretic properties, D-mannitol can also be used for medical treatment [21]. Usually, D-mannitol can be biosynthesized by enzymatic methods from D-fructose catalyzed by mannitol dehydrogenase (MDH) [18]. In D-allulose production, the reaction equilibrium between D-allulose and D-fructose is 30:70. The residual 70% D-allulose could be used for D-mannitol production.

The aim of this study is to sufficiently utilize D-fructose from fruit residues, and a two-stage biosynthesis strategy of valuable saccharides production was established. Firstly, KEase was used to produce D-allulose from fruit residues. Secondly, we produced D-mannitol from residual components using the whole-cells which co-express the formate dehydrogenase (FDH) and MDH. In order to reduce production cost and increase the efficiency of fruit waste disposal, fruit peels including pear peel, watermelon peel, Hami melon peel, orange peel, and mango peel were prepared to determine the production capacity of D-allulose and D-mannitol. After reaction, D-mannitol could be separated by cooling crystallization from the mixture of D-allulose and D-mannitol. This strategy provides a vital opportunity to advance the methodology of using D-fructose or low-cost biomass to biosynthesize high-value functional sugars. 

## 2. Materials and Methods

### 2.1. Strains, Plasmid, Reagents, and Materials

The ketose 3-epimerase (KEase) gene from *Dorea* sp. CAG317 (Dosp) (GenBank accession number: No. WP_ 022318236.1) was obtained from our previous study [22]. *E. coli* BL21(DE3) and *E. coli* DH5α strains and D-allulose and D-mannitol standards were purchased from Sangon Biotech Co., Ltd. (Shanghai, China). Sinopharm Chemicals Reagent (Shanghai, China) supplied the chemicals and reagents used in the purification of proteins and in the reaction process. The fruits including pear, orange, mango, watermelon, and Hami melon were purchased from the local market.

### 2.2. Heterogeneous Expression of KEase in E. coli

The plasmid pET-22b(+)-Dosp-KEase was transformed into *E. coli* BL21(DE3) for heterogeneous expression, and was grown in Luria-Bertani medium [23] supplemented with ampicillin (100 μg/mL), and the incubation condition was at 37 °C, 200 rpm. Isopropyl β-D-1-thiogalactopyranoside (IPTG), 1 mM was applied as an inducer when the culture indicator reached an OD_600_ of 0.6–0.8. After ITPG induction at 28 °C and 200 rpm for 6 h, the recombinant protein was overexpressed, and centrifugation was used to separate the cells at low temperature and 8000 rpm for 15–20 min. The cells were then washed with distilled water.

### 2.3. Purification of KEase

The cells were resuspended with 15 mL of lysis solution and sonicated for 15 min at 4 °C to disrupt them. After centrifuging the crude extracts at 8000 rpm at 4 °C for 15 min, the supernatant was filtered using a 0.22 μm aqueous phase Millipore filter. The enzyme was purified using a Ni^2+^ chelated Sepharose Fast Flow resin column after centrifugation and filtering. The column was first pre-equilibrated with the binding buffer, and then the supernatant went through the column. Afterward, the binding buffer was passed through the resin column to completely bind the protein with the resin column. The resin column was then washed with the washing buffer containing low concentration imidazole to elute undesired proteins. Ultimately, the elution buffer containing high concentration imidazole went through the resin column to collect the desired target enzyme. All of the formulations of the buffers referred to Zhang’s previous research [22]. To remove metal ions, the purified enzyme solution was dialyzed at 4 °C for 12 h in sodium phosphate buffer (PB, 50 mM, pH 6.0) containing 10 mM ethylenediaminetetraacetic acid (EDTA). It was then dialyzed against PB without EDTA (50 mM, pH 6.0) for 12 h at 4 °C twice. The protein concentration of the enzyme was determined by using the Bicinchoninic Acid (BCA) assay, and the bovine serum albumin (BSA) was used as a standard protein. 

### 2.4. Construction of the MDH-FDH Co-Expression System 

The D-mannitol 2-dehydrogenase (MDH) gene from *Caldicellulosiruptor morganii* Rt8.B8 (GenBank accession number: WP_045170409.1) and the formate dehydrogenase (FDH) gene from *Ogataea parapolymorpha* (GenBank accession number: EFW95288) were synthesized by Genewiz (Suzhou, China), and subcloned into the pETDuet-1 vector, between the *NcoI* and *EcoRI* restrict sites and *NdeI* and *XhoI* restrict sites, respectively. The expression method referred to 2.2 in this study. After inducing them with IPTG, the recombinant cells harboring pETDuet-1-MDH-FDH were centrifugated at low temperature and 8000 rpm for 15–20 min, followed by washing with distilled water. After discarding the supernatant, the bacteria cells are collected in the centrifuge tube. Whole-cell concentrations were measured spectrophotometrically (Shanghai Mapada Instruments Co., Ltd., Shanghai, China), and then the calculation was used to convert them to dry cell weight (DCW):DCW (g/L) = (0.4442 × OD600) − 0.021 [24]. 

### 2.5. Optimization of Whole-Cell Transformation Conditions

The effect of pH was examined at 35 °C with a pH range of 5.0–9.0. To investigate the maximum conversion yield of D-mannitol, three buffer systems were used: 50 mM NaHAc-HAc buffer (pH 5.0 to 6.0), 50 mM PBS buffer (pH 6.0 to 7.5), and 50 mM Tris-HCl buffer (pH 7.5 to 9.0). The effect of temperature was set at 50 mM PBS 6.5 and temperatures ranged from 30 to 50 °C. The impact of metal ions on the transformation efficiency was examined using eight metal ions (Co^2+^, Ni^2+^, Mg^2+^, Ca^2+^, Mn^2+^, Al^3+^, Zn^2+^ and Cu^2+^). The cell mass effects on conversion rate were investigated at four dry cell weight values of 0.4, 2, 4, 10 g/L using the PBS 6.5 buffer at 35 °C. Respectively, all the reaction solution was containing 40 g/L D-fructose, 40 g/L sodium formate and 0.2 mM NAD^+^, and all reaction was performed for 12 h then boiled for 10 min and centrifugated at 12,000 rpm for 5 min to stop the reaction. The D-mannitol formation was determined using high performance liquid chromatography (HPLC), and the conditions of the column and detector referred to Chen’s previous study [25]. Analysis conditions were as follows: mobile phase: ultrapure water consisting of 50 mg/L EDTA-Ca; running flow rate: 0.4 mL/min; column temperature: 85 °C; detector temperature: 30 °C. All samples were pretreated using Li’s method for HPLC analysis [26].

### 2.6. Preparation and Sugar Content Analysis of Fruit Residues

Fruit residues were washed and boiled for 20 min and then crushed with a juicer. The juice was obtained by filtration with muslin and centrifuged at 8000 rpm for 15 min, after which the supernatant was passed through a 0.22 μm aqueous phase Millipore filter. The sugar content of fruit residues was analyzed by HPLC. 

### 2.7. Production of D-Allulose and D-Mannitol from Fruit Residues

The pear peel juice was first adjusted to a pH of 6.5 with 1 mol/L NaOH, and then 0.1 μM Dosp-KEase purified enzyme and 1 mM MnCl_2_ were added into the pear peel juice. The reaction was performed at 60 °C for 3 h, and boiled for 10 min to terminate the reaction. The D-allulose formation was determined using HPLC. After D-allulose production, 2 g/L dry cell weight (DCW) whole-cell, 10 g/L sodium formate, 0.5 mM NAD^+^ were added into the system. The reaction was performed at 35 °C for 20 h. The reaction was stopped by boiling for 20 min and centrifugation at 12,000 rpm for 5 min. The D-mannitol formation was determined using HPLC. 

## 3. Results

### 3.1. Construction of Two-Stage Strategy Biosynthesis of D-Allulose and D-Mannitol 

In this study, we aim to utilize the D-fructose in fruit residue biomass to synthesize more valuable saccharides, D-allulose and D-mannitol (Figure 1). Usually, D-allulose and D-mannitol can be directly synthesized from D-fructose catalyzed by ketose 3-epimerase (KEase) and D-mannitol 2-dehydrogenase (MDH), respectively. It is known that around 30% D-fructose could convert into D-allulose, which is the major restriction of scale-up production of D-allulose. In comparison, D-mannitol enzymatic synthesis from D-fructose might theoretically approach a 100% conversion rate. This provides a solution to the issue that 70% D-fructose remained after D-allulose production. Meanwhile, MDH has strong specificity to D-fructose and could not react with D-allulose. Therefore, we constructed a novel approach for D-allulose and D-mannitol production in a two-stage system. In the first step, D-allulose was produced from D-fructose catalyzed by KEase, with a conversion yield of 30%. In the second step, the 70% residual D-fructose was catalyzed by MDH to synthesize D-mannitol. It should be mentioned that the substrate D-fructose was obtained from the fruit residues. 

In the production of D-mannitol, the dependency on cofactors (nicotinamide adenine dinucleotide (NADH) or nicotinamide adenine dinucleotide phosphate (NADPH)) for each MDH greatly reduces the yield of D-mannitol, and direct addition increases expenses. To address this issue economically, a technique for the reproduction of NADH or NADPH is required. Under this circumstance, a two-enzyme co-expression system can be employed for cofactor regeneration by converting two substrates into two products simultaneously, such as formate dehydrogenase (FDH) and glucose dehydrogenase (GDH), with formate and glucose as co-substrates, CO_2_ and gluconic acid as products. Therefore, FDH from *Ogataea parapolymorpha* was chosen to ligate to the MDH from *Caldicellulosiruptor morganii* Rt8.B8 to create a recombinant plasmid (pETDuet-1-MDH-FDH) and then to co-express in *E. coli* BL21(DE3), and the whole-cell was used in the reaction to catalyze D-fructose. A major advantage of FDH is that the byproduct CO_2_ can be easily separated from D-mannitol and is more applicable to our system. We successfully constructed the recombinant plasmid (pETDuet-1-MDH-FDH) and expressed these two proteins into *E. coli* BL21(DE3). The results of sodium dodecyl sulfate polyacrylamide gel electrophoresis (SDS-PAGE) indicated that the two proteins were fully expressed. As shown in Figure 2, the whole-cell protein band of approximately 36.0 kDa was consistent with the previous study of MDH, and another band of approximately 40.0 kDa was consistent with FDH.

### 3.2. Optimization of Whole-Cell Transformation Conditions

Although the optimal catalytic conditions of *C. morganii* MDH and *O. parapolymorpha* FDH were reported before, the optimal transformation conditions of whole-cell co-expressed these two enzymes had not been researched. In order to improve the D-mannitol yield, the whole-cell transformation conditions including pH, temperature, metal ions, and cell mass were optimized. It has been identified that pH 6.5 and 65 °C are the optimal conditions for the *O. parapolymorpha* FDH [27], while the *C. morganii* MDH had an optimal temperature of 75 °C and pH of 8.0 with D-fructose as substrate [28].

It can be seen from Figure 3 that the optimal temperature, pH, cell mass and metal ions were 35 °C, pH 6.5, 2 g/L DCW, and Mn^2+^. As shown in Figure 3a, in PBS buffer 6.0–7.5, the conversion rate was over 80%, and the optimum pH is 6.5, in which the conversion rate could reach 94.0%. From Figure 3b we can see that 35 °C brought the highest conversion yield, which is in the temperature range for the survival of *E. coli*. Although these two enzymes have higher activity at high temperature, for a lengthy reaction, cells would more stable and the conversion rate could reach a maximum value of 92% at 35 °C. It can be seen from Figure 3c that most metal ions had little effect on the conversion rate, which was similar to the control group without metal ions. In addition, Cu^2+^ significantly inhibited the transformation, and the conversion rate only reached 3.6%. The optimum metal ion was Mn^2+^, which was consistent with MDH. In the presence of Mn^2+^, the conversion rate reached 93.3% and was 24% higher than the control group that without metal ions. 

### 3.3. Sugar Content of Fruit Residues

The sugar content of the peel residue is an important indicator that determines how much value-added product we can produce, especially the content of D-fructose, which is the fundamental substrate for the bioproduction of D-allulose and D-mannitol in this strategy. After boiling, crushing, filtration and centrifugation, the water-soluble sugars in peel residues contain sucrose, D-glucose, and D-fructose. Figure 4 shows the sugar contents of fruit residues. For most fruit residues, D-fructose accounted for a high proportion, reaching 35–66%. And the content of D-fructose reached 16–40 g/kg of fresh residues, the lowest of which is orange peel and the highest of which is pear peel. The content of glucose was 7-25 g/kg, and the content of sucrose was 1–19 g/kg. It is noticeable that both contents of watermelon peel are the lowest. Among these fruit residues, pear peels showed the highest content of D-fructose, by approximately 40 g/kg, which indicated that pear peel residues represent a good prospect for the production of D-allulose and D-mannitol. In this study, pear peel was selected as the fruit residue to produce D-allulose and D-mannitol due to its high content of D-fructose.

### 3.4. Bioconversion of D-Allulose and D-Mannitol from Pear Peels

In this study, KEase from *Dorea* sp. CAG317 (Dosp-KEase) was used as a catalyst to produce D-allulose from the D-fructose of pear peels. The optimal pH and temperature for the Dosp-KEase have previously been determined to be pH 6.0 and 70 °C. Furthermore, it has the greatest enzyme activity of all known KEases (803 U/mg), and it also shows strong catalytic activity throughout a broad pH range of 5 to 8 [22]. As shown in Figure 5, 3.7 g/L D-allulose could be produced in 2 h at 60 °C and pH 6.5 when Mn^2+^ was present, with a conversion yield of 29.4%, and 9.41 g/L D-fructose remaining in the system, although the optimal metal ions for Dosp-KEase is Co^2+^, Mn^2+^ can also improve the conversion yield and the effect is similar to Co^2+^. Moreover, Mn^2+^ is the optimum metal ion in the transformation of D-mannitol, which avoided the addition of extra metal ions in the subsequent reaction. The conversion rate of D-allulose was 29%, which is very nearly the same as the maximum conversion rate of Dosp-KEase in D-fructose of 30%. By comparison, Song et al. reported that when using KEase from *A. tumefaciens* to convert D-fructose from Jerusalem artichoke into D-allulose, the conversion rate was 13.2%. Compared to conversion rates shown in the D-fructose reaction, the conversion rate was reduced by more than half, which might be caused by the high content of Fe^2+^ in Jerusalem artichoke [29]. Patel et al. also used fruit residues to biosynthesize D-allulose from pomace of apple and kinnow fruit by using covalently immobilized KEase that was environmental-friendly and recyclable, and the conversion rate was approximately 20% [30]. It is efficient to use Dosp-KEase to biosynthesize D-allulose from pear peel, and the conversion yield could reach 29.4%.

D-mannitol was typically produced by MDH reducing D-fructose at the C-2 position. In order to regenerate the cofactor NADH, we constructed a co-expression system to simultaneously express the FDH from *O. parapolymorpha* and MDH from *C. morganii* Rt8.B8 in *E. coli* BL21(DE3). The co-expressed whole-cell was used as a catalyst to biosynthesize D-mannitol from the residual D-fructose of pear peel juice. After D-allulose production, 2 g/L DCW whole-cell catalyst and 10 g/L co-substrate sodium formate were added into the reaction system. As shown in Figure 5, at pH 6.5 and 35 °C, D-fructose could convert into D-mannitol with a conversion yield of 93.5%. After 22 h, 8.79 g/L D-mannitol was eventually obtained from 9.41 g/L D-fructose. Kaup et al. reported using whole-cell co-expressed *L. pseudomesenteroides* ATCC 12,291 MDH with *M. vaccae* N10 FDH to bioproduce D-mannitol from D-fructose, and the conversion yield reached 73.0% [31]. However, most of the research on the bioproduction of D-mannitol from biomass has focused on the fermentation of lactic acid bacteria or yeast that usually requires additional nitrogen sources [32]. D-mannitol bioproduction by co-expressed whole-cells is easily operated and efficient, with a high conversion yield. 

### 3.5. Overall Mass Balance

According to the component analysis of each step, the mass balance diagram of each link is obtained, including epimerization, whole-cell catalysis, and separation steps. As shown in Figure 6, 1 kg fresh pear peel contains 39.0 g D-fructose, and after epimerization 10.8 g D-allulose was produced with a conversion rate of 29.4%. From the remaining 27.2 g D-fructose, 24.6 g D-mannitol was obtained through whole-cell catalysis with a conversion rate of 93.5%. It was an efficient strategy to take advantage of fruit and vegetable (FV) waste biomass that almost all D-fructose were converted to D-allulose and D-mannitol. 

A considerable amount of literature has been published on using low-cost biomass as raw material and KEases as catalysts for D-allulose production. As shown in Table 1, various FV residues and biomass were used to biosynthesize D-allulose, such as cane molasses, Jerusalem artichoke, inulin, jujube and FV residues. In order to produce D-allulose from FV wastes, Patel et al. employed the SUMO fusion of *A. tumefaciens* KEase as the catalyst, and the conversion yield reached 25–35% [31]. Yang et al. expressed 3 KEases in *C. glutamicum* and immobilized the cells to catalyze cane molasses to obtain D-allulose, finally obtaining 61.2 g/L D-allulose [32]. Li et al. constructed a two-enzyme system with KEase and exoinulinase, and obtained 21.4 g/L D-allulose from 100 g/L inulin in one pot [33]. Currently, the utilization of biomass for creating more value-added chemicals in addition to D-allulose has attracted increasing attention. Song et al. used *S. cerevisiae* to ferment residual D-fructose and biomass to produce bioethanol after the enzymatic production of D-allulose from biomass, finally obtaining 137.8 g D-allulose and 148.3 g bioethanol from 1 kg Jerusalem artichoke tubers dry mass, and 42.6 g D-allulose and 163.8 g bioethanol were produced from 1 kg cruciferous vegetable residues [29,34]. Sharma et al. used dextransucrase and SUMO fusion of KEase to catalyze 1 kg cane molasses, and finally produced 124 g prebiotic oligosaccharides and 37 g D-allulose [35]. Men et al. used D-glucose isomerase and KEase to produce D-allulose from jujube juice, and used *Pediococcus pentosaceus* PC-5 and *Lactobacillus plantarum* M to increase bioactivities and flavor volatiles; they obtained gamma-aminobutyric, branched-chain amino acids and other beneficial functional components besides D-allulose [36]. These studies are very practical in providing ideas for the production of D-allulose from biomass, and offer the potential for higher utilization efficiency, higher selectivity, lower energy costs, and the generation of fewer inhibitory byproducts. By comparison, in this study we realize the co-production of D-allulose and D-mannitol from D-fructose for the first time, offering a new idea for the full utilization of D-fructose. Furthermore, we applied this strategy to fruit residues. It is known that a growing number of fruit and vegetable wastes have led to a considerable burden on the environment in recent years. Using fruit and vegetable residues as a substrate for the bioproduction of D-allulose and D-mannitol provides a novel idea for the disposal of fruit and vegetable wastes from industrial processes, which produces great economic and social benefits, while also reducing negative environmental effects. Compared to other strategies that used cane molasses and Jerusalem artichoke as the raw materials, fruit residues are more convenient to handle without complicated processing. Furthermore, D-mannitol has a lower solubility of 18% (*w*/*v*) among most sugar alcohols [37], which has the potential to separate D-allulose and D-mannitol in industrial production.

## 4. Conclusions

In this study, a novel strategy was established to efficiently utilize D-fructose from a low-cost biomass for value-added saccharides bioproduction. The first stage involved using ketose 3-epimerase (KEase) to stimulate the production of D-allulose from the D-fructose of fruit peels. In the second step, an engineered strain *E. coli* BL21(DE3) simultaneously expressing D-mannitol 2-dehydrogenase (MDH) and formate dehydrogenase (FDH) was constructed and used as the whole-cell catalyst for the bioproduction of D-mannitol from the residual D-fructose. After optimization, about 29.4% of D-fructose pear peels could be converted to D-allulose, and almost all of the residual D-fructose was transformed into D-mannitol with a conversion yield of 93.5%. Finally, 10.8 g D-allulose and 24.6 g D-mannitol could be produced by 1 kg fresh pear peel. Furthermore, D-mannitol could easily be separated from D-allulose by cooling crystallization, which offers the potential for D-allulose and D-mannitol production. In summary, this strategy provided a vital alternative to use D-fructose or low-cost biomass to biosynthesize high-value functional sugars and a novel approach to separate D-allulose from the reaction system. Moreover, this strategy provides significant economic and social benefits, while also reducing negative environmental effects. 

## Figures and Tables

**Figure 1 foods-11-03613-f001:**
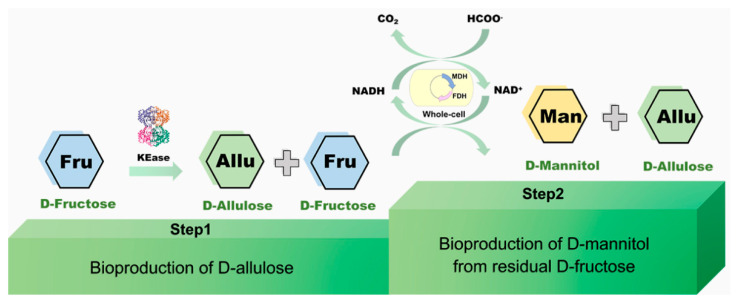
Schematic representation of bioproduction of D-allulose and D-mannitol from D-fructose via multi-enzyme and whole-cell catalysis system developed in this study.

**Figure 2 foods-11-03613-f002:**
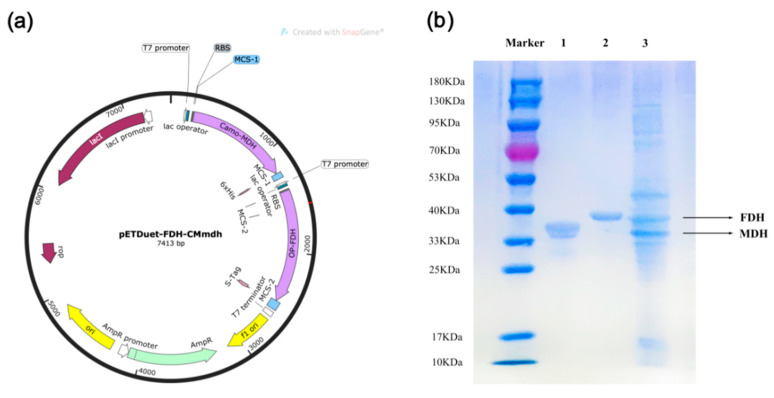
pETDuet-1-MDH-FDH recombinant plasmid construction map (**a**), and SDS-PAGE analysis of the purified enzymes and co-expression recombinant cells of MDH and FDH (**b**). Lane M, protein marker; lane 1, the purified enzyme of *E.coli* BL21(DE3)-pETDuet-1-MDH, lane 2, the purified enzyme of *E.coli* BL21(DE3)-pET22(b)-FDH, lane 3, the whole-cell of *E.coli* BL21(DE3)-pETDuet-1-MDH-FDH.

**Figure 3 foods-11-03613-f003:**
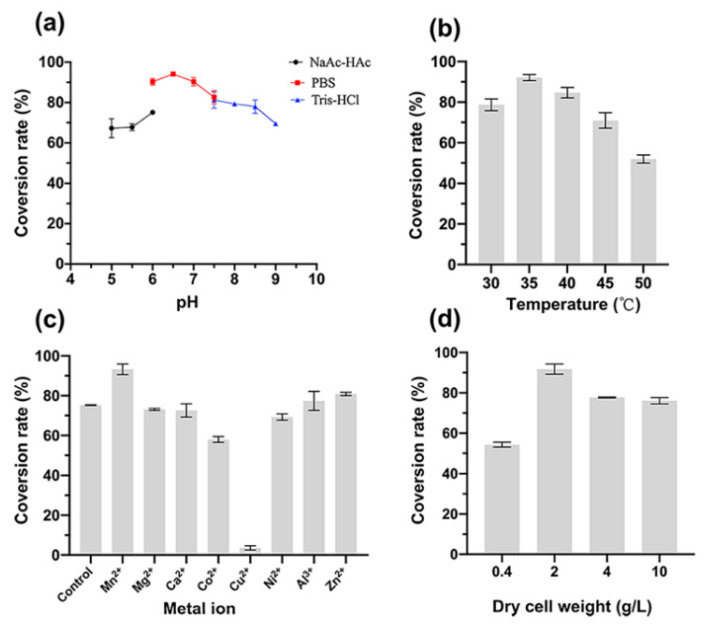
Effect of pH (**a**), temperature (**b**), metal ions (**c**), and dry cell weight (**d**) on the conversion rate of MDH-FDH whole-cell.

**Figure 4 foods-11-03613-f004:**
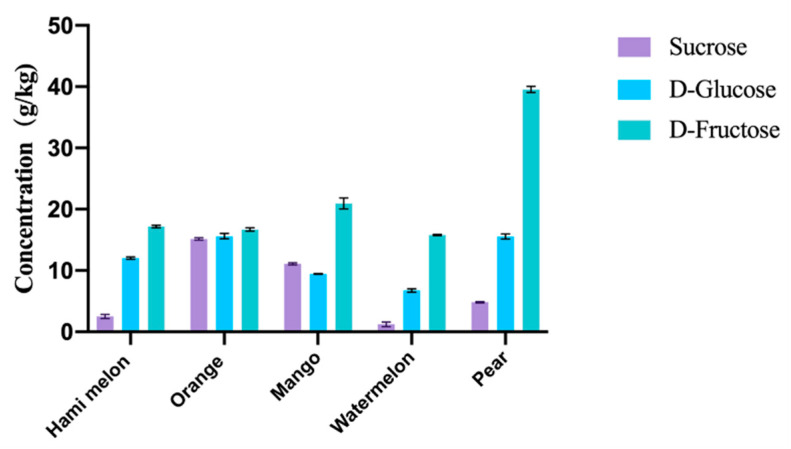
Sugar content of fruit peels after boiling and crushing pretreatment.

**Figure 5 foods-11-03613-f005:**
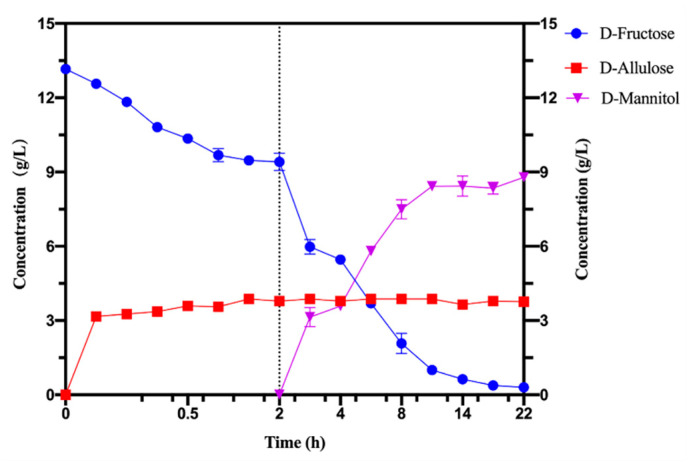
Bioconversion of pear peel into D-allulose and D-mannitol catalyzed by purified Dosp-KEase and MDH-FDH whole-cell with two-stage strategy.

**Figure 6 foods-11-03613-f006:**
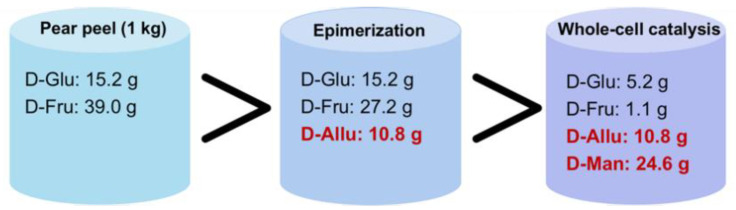
Overall mass balance for epimerization; whole-cell catalysis.

**Table 1 foods-11-03613-t001:** Bioproduction of D-allulose from biomass.

Raw Material	Method	Production	Reference
Fruit and vegetable residues	N-terminal SUMO fusion of *A. tumefaciens* KEase as the biocatalyst.	The conversion yield of D-allulose reached 25–35% in fruit and vegetable residues.	[33]
Cane molasses	Dextransucrase from *L. mesenteroides* MTCC 10,508 to produce prebiotic oligosaccharides, SUMO fusion of *A. tumefaciens* KEase to synthesize D-allulose.	124 g oligosaccharides (DP3-DP6) and 37 g D-allulose were obtained from 1 kg cane molasses.	[35]
Cruciferous vegetable residues	Residues was hydrolyzed into D-glucose and D-fructose by cellulase at first; then, D-glucose was fermented to bioethanol by yeast, while D-fructose was converted to D-allulose by KEase.	49.4 g D-allulose and 166.7 g bioethanol were obtained from 1 kg dry weight cabbage (320.0 g glucose, 142.0 g fructose), 42.6 g D-allulose and 163.8 g bioethanol were obtained after separation.	[34]
Jerusalem artichoke tubers	Jerusalem artichoke was first hydrolyzed into D-glucose and D-fructose by exoinulinase; then, D-glucose was fermented to bioethanol by yeast, while D-fructose was converted to D-allulose by KEase.	173.9 g D-allulose and 180.3 g bioethanol were obtained from 1 kg dry weight jerusalem artichoke (151.4 g glucose, 564.7 g fructose), 137.8 g D-allulose and 148.3 g bioethanol were obtained after separation.	[29]
Fruit residues	Using covalently immobilized KEase onto functionalized iron oxide magnetic nanoparticles as catalyst.	Immobilized enzyme is stable and still has 80% activity after 60 days of storage at 4 °C; and still has 90% activity after catalyzing ten cycles, and the conversion rate in kinnow and apple pomace is about 20%.	[30]
Inulin	A one-pot two-enzyme reaction system with *Dorea* sp. KEase and *Aspergillus piperis* exoinulinase.	21.4 g/L D-allulose was obtained from 100 g/L inulin.	[26]
Jerusalem artichoke	A one-pot two-enzyme reaction system with *Ruminococcus* sp. KEase and *Bacillus velezensis* exoinulinase.	The final high-fructose syrup contained 10.4, 29.2, and 10.3 g/L D-glucose, D-fructose, and D-allulose.	[38]
Cane molasses	Integrated expression of 3 KEases in *C. glutamicum and* immobilized the cells of KEases and invertase to catalyze cane molasses for two-step reaction.	61.2 g/L D-allulose was obtained from cane molasses (300.0 g/L sucrose, 16.5 g/L glucose, 69.0 g/L fructose).	[35]
Jujube	D-glucose isomerase and KEases converted D-glucose and D-fructose into D-allulose, *Pediococcus pentosaceus* PC-5 and *Lactobacillus plantarum* M were employed to increase bioactivities and flavor volatiles components.	110 g/L D-allulose was obtained from jujube juice (352 g/L glucose, 360 g/L fructose), 100 mg/L gamma-aminobutyric acid was obtained after fermentation.	[39]

## Data Availability

The data presented in this study are available on request from the corresponding author.

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
