# Peer review of "Efficient Utilization of Fruit Peels for the Bioproduction of D-Allulose and D-Mannitol"

_foods, 2022, doi:10.3390/foods11223613_

Round 1

Reviewer 1 Report

The article is adequate enough and may be considered provided the following comments are considered.

1.    Consider revising the title as “Efficient utilization of low-cost biomass resources for the bioproduction of D-allulose and D-mannitol

2.    Clearly elaborate the aim of study in the introduction section

3.    Extend the introduction section with citing more literature.

4.    Use the full form of all abbreviated terms at their first use.

5.    Emphasize the cost effectiveness, easiness and feasibility of this method over others.

6.    Overall English language is fine but still need some improvement

7.    Try to bring similarity down to 19% or below as per recommended criteria of journal and respective institution.

Reviewer 2 Report

This research reported a methodology for producing D-allulose and D-mannitol from low-cost biomass resources. The reviewer has some comments that the authors may want to take into consideration.

1. The abstract and introduction parts are confusing. In the abstract, the authors stated that fruit and vegetable materials were used as raw materials. In the manuscript, only the results from several fruits were reported. Without clarification, it was concluded that the best resource for the synthesized sugar was pear (peel). It seems that the title should be revised to reflect the context of the manuscript.       

2. The details of the fruit samples used should be provided. Some physicochemical properties of the fruit parts should be provided. This could explain why the pear peel was the best material. Do the mangoes from different varieties result in different results?

3. It was not clear how the optimization was conducted. There were a number of factors reported. Was the optimization outcome based on the design of experiments?

4. There are also some minor issues that need to be double-checked for grammatical problems.   

Round 2

Reviewer 2 Report

The revised manuscript is ready for official publication, so I think it can be accepted.

Author Response

Comment 1:

The revised manuscript is ready for official publication, so I think it can be accepted.

Response: 

  We sincerely thank the reviewer for the professional review work on our article and the valuable feedback that helped us to improve the quality of our manuscript. And we appreciate the reviewer for giving positive comments and valuable suggestions.